# *Helicobacter pylori* Across Continents: Contrasts in Epidemiology, Genetics, Clinical Impact, and Management Between East and West

**DOI:** 10.3390/ijms262311408

**Published:** 2025-11-25

**Authors:** Ken Namikawa, Fehima L. Purisevic, Jon B. Thorsteinsson, Einar S. Bjornsson

**Affiliations:** 1Division of Gastroenterology and Hepatology, Department of Internal Medicine, Landspitali University Hospital, 101 Reykjavik, Iceland; 2Department of Gastroenterology, Cancer Institute Hospital, Japanese Foundation for Cancer Research, Tokyo 135-8550, Japan; 3Faculty of Medicine, University of Iceland, 101 Reykjavik, Iceland

**Keywords:** *Helicobacter pylori*, geographic variation, east and west, diagnostic and treatment strategies, antibiotic resistance

## Abstract

*Helicobacter pylori* (*H. pylori*) remains one of the most widespread and clinically significant bacterial infections globally, affecting over half the world’s population. This review explores the geographic contrasts in *H. pylori* epidemiology, genetic diversity, disease burden, and management strategies, with a particular focus on differences between Eastern and Western regions. East Asia bears a disproportionately high burden of *H. pylori*-associated diseases, especially gastric cancer, due to more virulent strains and distinct patterns of gastritis. Genetic variations in key virulence factors such as *cagA* and *vacA* contribute to regional differences in clinical outcomes. Diagnostic and therapeutic approaches also vary widely, shaped by healthcare infrastructure, screening practices, and antibiotic-resistance profiles. Advances in endoscopic techniques and personalized medicine have improved early detection and treatment in high-incidence regions, while Western countries face challenges in implementing widespread screening due to lower disease prevalence. This review highlights the importance of region-specific strategies, genomic surveillance, and international collaboration to address disparities in disease outcomes and improve access to care. Understanding the complex interplay between bacterial genetics, host factors, and environmental influences is essential for developing effective prevention, diagnostic, and eradication programs tailored to diverse populations.

## 1. Introduction

*Helicobacter pylori* (*H. pylori*) is a spiral-shaped, Gram-negative bacterium that colonizes the human stomach and infects more than half of the global population [1]. Since its discovery by Marshall and Warren in 1982 [2], *H. pylori* has been recognized as a major contributor to gastrointestinal diseases, including peptic ulcer disease, mucosa-associated lymphoid tissue (MALT) lymphoma, and gastric cancer [3]. Its classification as a Group 1 carcinogen by the World Health Organization (WHO) underscores its clinical significance [4].

Although the global prevalence of *H. pylori* has declined in some regions due to improved sanitation and the widespread adoption of eradication therapy, it remains a major public health concern [5]. Gastric cancer continues to pose a significant global burden [6], and a large-scale analysis by de Martel et al., based on data from the WHO and the International Agency for Research on Cancer (IARC), revealed that *H. pylori* is the leading infectious cause of cancer worldwide. It accounts for the highest number of cancer cases among all infection-related malignancies—primarily gastric cancer—and represents more than one-third of all cancers attributable to infectious agents [7].

In addition to its role in gastric carcinogenesis, *H. pylori* has been linked to extra-gastric conditions such as iron-deficiency anemia and idiopathic thrombocytopenic purpura, further broadening its clinical impact [8]. Notably, the burden of *H. pylori*-related disease is disproportionately concentrated in the Eastern part of the hemisphere, where both infection rates and gastric cancer incidence are significantly higher than in Western countries [9]. East Asian strains tend to be more virulent and more closely associated with gastric cancer than Western strains [10], and this difference is regarded as one of the underlying causes of the geographic variation in gastric cancer incidence. Diagnostic methods, treatment protocols, and clinical guidelines also vary widely across regions, shaped by differences in healthcare infrastructure and antibiotic-resistance patterns [11]. Moreover, rising antibiotic resistance continues to challenge the efficacy of current treatment strategies [12].

These findings underscore the need for region-specific approaches to screening, prevention, and treatment. Therefore, this narrative review aims to synthesize current evidence on the geographic divergence of *H. pylori*, including the most recent research, with a particular focus on the East—defined here as Asia and the Middle East, with special attention paid to East Asia (e.g., Japan, China, and South Korea)—and the West, including Europe and the Americas, especially North America (e.g., Germany, France, the United Kingdom, the United States (USA), and Canada).

It examines six key domains: epidemiology and prevalence, transmission and reinfection, genetic diversity and population structure, clinical manifestations and disease risk, diagnostic approaches, and regional differences in treatment strategies and antibiotic resistance. By comparing these aspects across regions, this review highlights critical disparities and offers timely insights into the evolving landscape of *H. pylori* research and clinical management.

## 2. Epidemiology and Prevalence

Based on regional prevalence estimates, there were approximately 4.4 billion individuals with *H. pylori* infection worldwide in 2015, or around 60% of the world’s population [1]. However, it can be challenging to accurately estimate the prevalence of *H. pylori* in the general population as the sensitivity and specificity of the different diagnostic modalities are quite variable. Some diagnostic modalities can detect active infection, whilst others, such as serological tests of *H. pylori* antibodies, cannot differentiate between active or previous infection [13]. This is particularly relevant for patients who have undergone eradication therapy for *H. pylori*.

Prevalence is highly variable between Eastern and Western regions. In East Asia, the prevalence in Japan, Korea, and China has been decreasing, with an estimated prevalence of 52% [1], 54% [1], and 56% [1], respectively. In Mongolia, the prevalence has been reported as 74% [14]. Notably, Japan has experienced a marked decline in *H. pylori* prevalence, particularly among younger populations. Ueda et al. demonstrated that infection was most common among individuals born between 1940 and 1949, with the risk decreasing by approximately 15% for every successive 10-year birth cohort [15]. A separate meta-regression analysis of approximately 170,000 individuals further showed a decline from 67% among those born in 1930 to 35% in those born in 1970, declining further to approximately 7% in those born in 2000 [16].

In contrast, Western countries report lower prevalence overall. In Iceland, the prevalence of *H. pylori* based on serological antibodies among a random sample of individuals, 24–50 years old, showed a prevalence of 36% [17], while in Switzerland, the estimated prevalence was 19% [1]. In the US, the prevalence of *H. pylori* was 36% in the general population. However, substantial variation in *H. pylori* prevalence exists even within the same continent, showing a 75% prevalence among the Alaskan Indigenous population according to a recent systematic review and meta-analysis [1]. Prevalence also differs within the same country or region among different races and ethnicities; in the United States, the prevalence is substantially higher in non-White races and ethnicities [18,19]. In Europe, research suggests that *H. pylori* remains highly prevalent in migrant communities [20]. A nationwide study of United States veterans reported a decline in *H. pylori* prevalence, from 36% in 1999–2006 to 18% in 2013–2018 [21]. This reduction was observed across all racial and ethnic groups; however, the disproportionate burden of infection among non-Hispanic Black and Hispanic individuals compared with non-Hispanic White individuals persisted. Demographic variables accounted for approximately 5% of the variation in *H. pylori* positivity, with race and ethnicity contributing the most significant proportion [21].

Beyond the East and West, substantial variation in prevalence is observed globally. A systematic review and meta-analysis by Hooi et al. showed that the highest prevalence of *H. pylori* was found in Africa (79%), Latin America, and the Caribbean (63%), and the lowest prevalence in Oceania (24%) [1]. As an example, Nigeria and Libya have reported prevalence rates of 88% and 76%, respectively. In Latin America and the Caribbean, prevalence has remained relatively stable, with 63% before 2000 and 60% after 2000 [1]. In Oceania, prevalence declined from 27% to 19% after 2000.

These findings illustrate the considerable disparities in infection rates between populations and indicate that the prevalence of *H. pylori* is influenced, to a large extent, by national socioeconomic conditions. Factors such as access to clean water, healthcare availability, modern sanitation, and rising antibiotic resistance further contribute to these differences [20].

Globally, the prevalence of *H. pylori* has declined over time. This reduction is likely largely due to decreased transmission resulting from improved sanitation and living conditions in recent decades, with the most pronounced decline occurring during periods of industrialization [1]. Therefore, in many countries, prevalence continues to increase with age, reflecting historical exposure patterns and cohort effects.

Additional contributing factors include the widespread availability of effective treatment, as well as population-based testing and eradication programs in countries with high gastric cancer incidence [22,23]. A meta-analysis published in 2023 by Yunhao Li et al. [5] examined prevalence trends of *H. pylori* infection between 1980 and 2022 and confirmed a global decline during this period, particularly between 2011 and 2022 [5]. Although the prevalence of *H. pylori* infection is decreasing in many countries, it remains high in most developing nations. After 2000, the prevalence of *H. pylori* infection declined in several regions: in Europe, it dropped from 49% to 40%; in North America, from 43% to 27%; and in Oceania, from 27% to 19%. In contrast, *H. pylori* prevalence remained relatively stable in Asia—53.6% before 2000 and 54.3% after 2000—and in Latin America and the Caribbean, rates were at 63% before 2000 and 60% after 2000 [1]. A nationwide study of United States veterans reported a decline in *H. pylori* prevalence, from 36% in 1999–2006 to 18% in 2013–2018 [21]. This reduction was observed across all racial and ethnic groups; however, the disproportionate burden of infection among non-Hispanic Black and Hispanic individuals compared with non-Hispanic White individuals persisted. Demographic variables accounted for approximately 5% of the variation in *H. pylori* positivity, with race and ethnicity contributing the most significant proportion [21]. In many countries, the prevalence of *H. pylori* infection increases with age. Notably, Japan has experienced a marked decline in *H. pylori* prevalence, particularly among younger populations. Ueda et al. demonstrated that the infection was most common among individuals born between 1940 and 1949, with the risk of infection decreasing by approximately 15% for every successive 10-year birth cohort [15]. Another study from Japan, based on a systematic review and meta-regression analysis of around 170,000 individuals, found that the predicted prevalence of *H. pylori* infection declined across birth cohorts—from 67% among those born in 1930 to 35% in those born in 1970, declining further to approximately 7% in those born in 2000 [16]. Additionally, one study showed that the aggregate prevalence of *Helicobacter pylori* infection among urban Chinese residents was 27.08% [24], representing a substantial decrease compared with the earlier estimated prevalence of 44.20% for the broader Chinese population [25].

## 3. Transmission and Reinfection

*H. pylori* transmission and recurrence vary by region, shaped by biological, environmental, and social factors. Evidence supports the role of multiple transmission routes—such as gastro-oral, oral–oral, and fecal–oral (Figure 1)—though their relative importance remains debated.

Reinfection patterns and public health measures, including water, food, and intrafamilial exposure, reflect broader socioeconomic disparities.

### 3.1. Gastro-Oral Transmission

*H. pylori* can be transmitted through exposure to the stomach content of infected individuals. Iatrogenic transmission has been detected due to exposure to contaminated endoscopic equipment [26]. More relevant to community transmission, however, is infection through exposure to the vomitus of infected individuals. *H. pylori* has been cultured in high quantities from vomitus and shown to be transcriptionally active [27,28]. Additionally, individuals exposed to household members with gastroenteritis, particularly with vomiting or those with a sibling’s history of vomiting, are at greater risk of infection [29,30]. These findings suggest that infected individuals are uniquely contagious while experiencing gastroenteritis.

### 3.2. Oral–Oral Transmission

Although *H. pylori* has been detected in the oral cavity, its ability to colonize the site and its role in gastric infection remain controversial. Behaviors involving saliva exchange, such as sharing utensils [31] and maternal premastication [32], have been associated with increased infection risk. However, the correlation between oral and gastric *H. pylori* has been weak or inconsistent [33]. A complicating factor in studying this transmission route is the lack of standardized detection methods, which limits comparability across studies [34]. Oral *H. pylori* may also be detected because of contamination via gastroesophageal reflux or ingestion of contaminated food and drink [35]. Currently, the role of oral *H. pylori*, whether as a contaminant, reservoir, or transmission source, remains unclear.

### 3.3. Fecal–Oral Transmission

Although challenging, successful culture of *H. pylori* from feces has been reported in select studies, with the organism primarily existing in a viable but non-culturable (VBNC) state [36]. However, unlike oral detection, fecal *H. pylori* can be reliably detected using methods such as the SAT [37]. Using PCR assays, Momtaz et al. detected *H. pylori* in samples from gastric biopsy and stool at similar rates (78% vs. 72%), and found 93% strain concordance between the two sites, concluding that transmission likely occurs through the fecal–oral route [38]. Reports of coinfection with *H. pylori* and pathogens known to be transmitted through the fecal–oral route, such as hepatitis A virus and *Giardia duodenalis*, suggest a shared transmission pathway [39,40]. Additionally, a recent study detected an identical *H. pylori* strain in a pet owner and their dogs [41], as well as the presence of *H. pylori* in the feces of pet and farm animals [42]. These findings suggest that animal waste may serve as an additional reservoir for fecal–oral transmission to humans.

### 3.4. Waterborne Transmission

As contaminated fecal matter likely serves as a vehicle for transmission, water may act as an intermediary, particularly in settings with poor sanitation. Epidemiological studies have shown that individuals who rely on external water sources, such as river water [43] or wells [44], have a higher risk of infection compared to those using tap water. Supporting this association, viable *H. pylori* has been detected—and, in rare cases, successfully cultured—from seawater (in association with zooplankton), drinking water (after enrichment), and, most definitively, wastewater, from which multiple viable *H. pylori* isolates harboring virulent genotypes were obtained [45,46,47]. Experimental studies further support waterborne transmission. In a murine model, Boehnke et al. demonstrated dose-dependent gastric colonization following ingestion of *H. pylori*-contaminated water (SS1 strain), providing direct evidence of infectivity via this route [48]. However, a follow-up study was unable to demonstrate infectivity following ingestion of the VBNC form of SS1, possibly due to a lack of strain-specific infectivity or a difference in host susceptibility [49].

### 3.5. Foodborne Transmission

Foodborne transmission is also plausible. Consumption of raw vegetables has been associated with an increased infection risk, with contaminated irrigation water presumed to be the source of the contamination [50]. *H. pylori* can survive for several days in various foodstuffs, including milk, ground beef, lettuce, and carrots, with longer survival observed under refrigeration and in sterile or pasteurized products where competing microflora are absent [51]. A zoonotic transmission pathway has been proposed [33]. Pathogenic strains of *H. pylori* have been isolated and cultured from the raw milk and meat of livestock, including cows, sheep, and goats [52]. Momtaz et al. reported 93–99% sequence homology between *H. pylori* isolates from gastric biopsies from sheep and humans, suggesting that sheep may serve as a reservoir for human-infecting strains [53]. Further epidemiological evidence supports this hypothesis. In a study conducted in Poland, *H. pylori* prevalence was markedly higher among shepherds (98%) and their families (86%) compared to controls without animal contact (65%) [54]. Although direct transmission of *H. pylori* between animals and humans has not been conclusively demonstrated, the close genetic relatedness of strains and elevated infection rates among individuals with animal exposure provide indirect evidence supporting a zoonotic component in *H. pylori* transmission.

### 3.6. Familial vs. Horizontal Transmission

The aforementioned indirect transmission routes facilitate *H. pylori* spread beyond the immediate family—a pattern characteristic of rural areas and developing countries. In contrast, transmission in urban and developed settings is typically confined to close familial contact, likely via the gastro-oral or fecal–oral route, with maternal and sibling infection being the primary sources of spread [55,56]. Concordance of *H. pylori* strains between mothers and their children, as well as among siblings, supports these findings [57]. Risk is further elevated with a higher sibling count, a small age gap between siblings, and higher birth order [58]. However, maternal transmission seems to play the most significant role. A systematic review found no independent association between infected siblings and childhood infection once maternal infection status was adjusted for [59]. A five-year follow-up study in Japan supports this [60]. Infants born to seropositive mothers—who later tested positive—carried strains with DNA fingerprinting patterns identical to those of their mothers, strongly suggesting mother-to-child transmission as a key pathway of familial spread [60].

In rural settings and developing countries, *H. pylori* transmission appears more complex, with extrafamilial horizontal spread playing a major role. Schwarz et al. analyzed the genetic relatedness of *H. pylori* strains among family members in both rural South Africa and urban international settings in the UK, USA, Korea, and Colombia. Compared to their urban counterparts, rural families showed greater strain diversity, lower concordance among household members, and no association between kinship and genotypic similarity [61]. These patterns likely reflect transmission in the backdrop of inadequate sanitation infrastructure, where exposure to contaminated water and food, non-parental caregivers, and close contact among unrelated children contributes to a wider array of infection sources [55].

### 3.7. Recurrence

The relative contribution of different transmission routes is further reflected in *H. pylori* recurrence rates, which vary by setting. A systematic review with meta-analysis by Hu et al. reported a global annual recurrence rate of 4.3% in adults, which remained stable over the 27-year study period. Recurrence correlated positively with *H. pylori* prevalence (2% to 11%) and negatively with socioeconomic development according to the human development index (3% to 11%) [62]. In contrast, children had a higher annual recurrence of 13% [63].

Recurrence after confirmed eradication is caused either by reinfection or recrudescence, the latter defined as the reappearance of the original strain missed by confirmatory testing. Genotyping helps distinguish recrudescence, which involves the same strain, from reinfection, which involves a different strain, though this method is rarely used in clinical practice [63,64,65]. Determining their relative influence is important as recrudescence indicates suboptimal treatment, whereas reinfection reflects ongoing transmission. Recurrence risk is higher within 24 months of eradication (9% vs. 2%) and with low-efficacy regimens (11% vs. 5%), supporting recrudescence as the main cause of early recurrence [66,67].

In a nested meta-analysis, Niiv et al. reported similar annual recurrence rates during the first and second year after eradication in developing countries (13% vs. 12%), but decreasing rates in developed countries (2.7% vs. 1.5%), supporting the predominance of reinfection in developing countries [68]. In contrast, recrudescence appears to predominate in developed countries. Studies from Australia, The Netherlands, and the UK found identical strains before and after eradication therapy, implicating recrudescence [69]. In Japan, two studies nearly a decade apart (2003, 2012) reported annual reinfection rates of 2% and 0.22%, respectively [64,65]. This decline coincided with lower *H. pylori* prevalence and improved sanitation, indicating a shift towards recrudescence [65]. The low recurrence rate observed after successful *H. pylori* eradication, particularly in sanitary, low-prevalence settings, translates into durable clinical benefit, with corresponding reductions in gastric cancer incidence and peptic ulcer recurrence [70,71].

### 3.8. East–West Differences in H. pylori Transmission and Recurrence

Previous reports suggest that *H. pylori* transmission patterns and recurrence rates differ between Asian and Western countries. In rural settings, which characterize many parts of Asia, an extrafamilial horizontal pattern of transmission has been reported [61], whereas in more urbanized and developed settings, typical of much of the West, familial transmission appears to predominate [61]. With respect to recurrence, Hu et al. reported low annual recurrence rates in Europe (2.7%) and Oceania (2.0%) compared to Asia (5.6%) [62]. In contrast, recurrence rates in North America (6.2%) and South America (7.7%) were also relatively high [62]. Importantly, there was substantial heterogeneity within each continent (I^2^ > 80%), indicating that recurrence rates varied widely across studies, even within the same region [62].

However, there are no studies that directly compare transmission routes or recurrence rates between East and West using the same methodology. Additionally, there is no evidence that differences in infectivity are due to specific *H. pylori* strains. Therefore, based on current knowledge, these differences in transmission and recurrence are thought to result from changes in sanitation and living conditions associated with urbanization, rather than from bacterial strain differences.

## 4. Genetic Diversity and Population Structure

*H. pylori* is one of the most genetically diverse bacterial species known, with extensive variation in its genome driven by recombination, mutation, and horizontal gene transfer [72,73,74]. This diversity is not random—it reflects the bacterium’s co-evolution with human populations and has significant implications for virulence and disease outcomes.

### 4.1. Global Phylogeography and Lineage Diversity

Initial phylogenetic studies using multilocus sequence typing (MLST) identified four major *H. pylori* populations: hpEurope, hpEastAsia, hpAfrica1, and hpAfrica2 [75]. Subsequent research expanded this framework to include additional subpopulations such as hpAsia2, hpSahul, and hpAmerind [76]. More recently, a global analysis of complete *H. pylori* genomes has reported novel subpopulations, further refining this classification and revealing complex patterns of recombination and gene flow across continents [74]. These lineages mirror ancient human migration patterns and are associated with specific geographic regions. For instance, hpEastAsia strains dominate in China, Japan, and Korea, while hpEurope strains are prevalent in Western countries [74].

Intra-national population structure also reflects geographic and ethnic diversity. For example, in China, hpEastAsia strains exhibit regional clustering and genetic diversity, reflecting internal migration and ethnic variation [77]. These patterns highlight the importance of localized genomic surveillance and belong within the broader context of geographic subpopulation analysis.

### 4.2. Virulence Gene Profiles

The cytotoxin-associated gene A (*cagA*) [78,79,80,81], which encodes a cytotoxin-associated protein that disrupts host cell signaling and promotes carcinogenesis [82], is a key virulence factor in *H. pylori*. The prevalence of *cagA*-positive strains is significantly higher in East Asia compared to Western countries [83]. The evidence from multiple studies demonstrates that nearly all *H. pylori* strains in East Asia are CagA (*cagA* gene product)-positive, while the prevalence of CagA-positive strains in Western countries is approximately 60% among *H. pylori* isolates, as supported by molecular and epidemiological data [10,84,85].

East Asian strains are particularly notable for their virulence. Importantly, the Glu-Pro-Ile-Tyr-Ala (EPIYA)-D phosphorylation motif found in East Asian *cagA* variants is more oncogenic than the EPIYA-C motif common in Western strains [83].

The prevalence of East Asian-type CagA among *H. pylori* strains is characteristically high in East Asia, with approximately 90% of isolates in mainland Japan, China, and Korea carrying the East Asian-type CagA genotype [84,86,87]. In contrast to these countries, where East Asian-type CagA predominates, regional variation is evident across Asia. In Okinawa, located in southern Japan, the prevalence of East Asian-type cagA is notably lower at 70.3% [88]. In the Philippines, all *H. pylori* strains examined in a recent cohort were CagA-positive, yet only 26.3% were classified as the East Asian type [89]. In Thailand, while the overall CagA positivity rate reaches 93.2%, just 11.6% of these strains carry the East Asian-type CagA, with the majority belonging to the Western-type lineage [90].

The vacuolating cytotoxin gene (*vacA*) [78,91], which encodes a vacuolating cytotoxin, also exhibits geographic variation. The s1/m1 genotype, associated with higher cytotoxicity, is more prevalent in East Asia, while the less virulent s2/m2 genotype is common in Western populations [92,93]. These differences contribute to the higher incidence of gastric cancer in East Asia.

Other virulence factors, such as blood group antigen-binding adhesin (*babA*), outer inflammatory protein A (*oipA*), *iceA* (induced by contact with epithelium), and duodenal ulcer-promoting gene A (*dupA*), show regional variability. For instance, *babA*, which facilitates adhesion to gastric epithelial cells, is generally more prevalent in East Asian strains [94], contributing to their higher virulence profiles [10], although it is also present in Western strains associated with severe disease [93]. Functional *oipA*, which enhances inflammation, is more common in East Asian strains [10]. The status of *iceA* exhibits marked geographic variation among *H. pylori* strains; however, neither *iceA* alone nor combinations of *iceA*, *vacA*, and *cagA* are reliable predictors of clinical outcomes such as gastric cancer, peptic ulcer, or atrophic gastritis, as consistently demonstrated by international collaborative studies [95].

### 4.3. Microevolution and Genomic Plasticity

Recent genomic studies have revealed that *H. pylori* undergoes rapid microevolution within individual hosts, driven by recombination, mutation, and selective pressures such as host immunity and antibiotic treatment [74,96,97]. *H. pylori* evolves at a markedly higher rate than most enteric bacteria. Comparative genomic analyses have estimated the synonymous substitution rate of *H. pylori* at approximately 1.38 × 10^−5^ substitutions per site per year [98], based on intra-host comparisons across multiple isolates. Longitudinal sampling over a three-year period yielded an upper bound of 4.1 × 10^−5^ [99]. Consistent with these findings, Kennemann et al. analyzed sequential isolates from chronically infected individuals and reported a mutation rate of approximately 2.5 × 10^−5^ substitutions per site per year [97]. Moreover, its recombination-to-mutation ratio (r/μ) has been estimated at ≥1 in multiple studies, with reported values of ~6.9 (86), ~9.5 [98], and ~14.5 [96], indicating that recombination contributes more to sequence diversification than mutation. Compared to other enteric bacteria, *H. pylori* exhibits markedly higher rates of both mutation and recombination.

In contrast, Escherichia coli evolves much more slowly, at ~4.5 × 10^−9^ substitutions per site per year [100], and exhibits a strongly clonal population structure with minimal recombination, as evidenced by the presence of stable “clonal frames” across strains [101]. These features highlight *H. pylori*’s exceptional plasticity and its capacity for rapid intra-host adaptation, with direct implications for persistence, immune evasion, and antibiotic resistance. This intra-host diversification leads to the emergence of genetically distinct subpopulations over time, complicating diagnosis and treatment.

East Asian strains, in particular, exhibit elevated recombination rates and genomic plasticity, which may enhance their adaptability and persistence in the gastric environment [74,96]. This is exemplified by increased allelic diversity in key virulence-associated loci such as *cagA*, *vacA*, *babA*, and *sabA*, which modulate host cell signaling, immune evasion, and epithelial adherence [74]. Comparative genomic studies have also revealed frequent recombination events within outer membrane protein families and restriction–modification systems, contributing to strain-specific host interactions and niche adaptation [96]. These dynamic evolutionary processes contribute to strain-specific virulence and resistance profiles.

Global genome analyses have also identified recombination hotspots and gene flow pathways, offering insights into how *H. pylori* evolves in response to environmental and host factors [74,96]. These mechanisms may also accelerate the emergence and spread of antibiotic resistance, as natural transformation and homologous recombination enable the uptake and incorporation of resistance-conferring mutations [102,103]—such as those in the *23S rRNA* gene (clarithromycin) [104,105], *pbp1A* (amoxicillin) [106], and *rpoB* (rifampicin) [107,108]. This evolutionary flexibility underscores the need for personalized therapeutic strategies based on strain-level genomic data.

Understanding these microevolutionary mechanisms is essential for developing effective diagnostics, treatments, and surveillance systems tailored to regional and individual variations.

### 4.4. Clinical Implications

The genetic makeup of *H. pylori* strains influences clinical outcomes. East Asian strains are more strongly associated with gastric adenocarcinoma, while Western strains are more often linked to duodenal ulcers [109]. These clinical differences reflect distinct inflammatory patterns: East Asian strains typically induce pan-gastric atrophy, leading to reduced acid secretion and a lower risk of duodenal ulceration, whereas Western strains tend to cause antral-predominant gastritis, resulting in increased acid output and a higher incidence of duodenal ulcers [10,110,111]. Such pathophysiological divergence has direct implications for screening strategies as populations with a higher risk of gastric cancer—such as those in East Asia—have adopted routine endoscopic surveillance programs, while duodenal ulcer-prone populations may benefit more from symptom-based approaches [112,113].

The development of vaccines targeting *H. pylori* has faced significant challenges. Genetic variability, including differences in key antigens such as *cagA* and *vacA* across populations, complicates the development of a universal vaccine [114]. Understanding regional differences in genetic profiles is essential for region-specific diagnostic and therapeutic strategies, the development of diagnostic tools tailored to diverse populations, and guiding future vaccine research.

## 5. Disease Burden and Clinical Patterns

The clinical consequences of *H. pylori* infection span a wide spectrum of gastrointestinal and extra-gastrointestinal diseases. These outcomes are shaped not only by bacterial virulence factors, as discussed in Section 4, but also by host genetics, environmental exposures, and healthcare practices. Importantly, the burden and presentation of *H. pylori*-associated diseases differ significantly between Eastern and Western populations, necessitating region-specific diagnostic and therapeutic strategies [115,116].

### 5.1. Peptic Ulcer Disease

Peptic ulcer disease remains one of the most common manifestations of *H. pylori* infection. In Western countries, duodenal ulcers are more prevalent, whereas gastric ulcers dominate in East Asia [109]. This divergence is partly explained by strain-specific virulence genes: East Asian strains more frequently harbor *cagA* with EPIYA-D motifs and *vacA* s1/m1 alleles, which induce stronger mucosal damage in the stomach [83]. These strain-specific virulence profiles contribute not only to ulcer localization but also to the severity and progression of mucosal injury. Comparative genomic and epidemiological studies have demonstrated that East Asian strains exhibit enhanced inflammatory signaling and epithelial disruption, whereas Western strains tend to be less virulent, often associated with antral-predominant gastritis and duodenal ulcers [117,118,119,120,121].

Host factors also contribute. Polymorphisms in pro-inflammatory cytokines such as IL-1β and TNF-α have been associated with increased gastric cancer risk through enhanced inflammatory responses and mucosal injury [122,123,124,125]. Although some studies suggest that these high-risk genotypes may be more prevalent in East Asian populations [126], research from Western countries has also demonstrated the presence of pro-inflammatory variants within their populations [127]. These findings highlight the complexity of host genetic contributions to disease patterns and underscore the need for region-specific risk stratification.

### 5.2. Gastric Cancer

*Gastric adenocarcinoma is the most severe outcome of chronic H. pylori infection.* Major pathoway for this progression is well illustrated by the Correa’s cascade, consisting of a sequential progression from chronic non-atrophic gastritis to atrophic gastritis, intestinal metaplasia, and dysplasia, which is a well-recognized model of development of non-cardia gastric cancer [128].

East Asia—particularly Japan, Korea, and China—has among the highest gastric cancer incidence rates globally, as reported by GLOBOCAN 2022 [129]. In contrast, Western countries report significantly lower gastric cancer rates, despite comparable *H. pylori* infection prevalence in some regions [1].

Recent studies have reported regional differences in the mutation spectrum of gastric cancer, suggesting that the carcinogenic pathways of gastric cancer may vary between Eastern and Western populations. In particular, the diversified mutation patterns observed in Eastern Europe imply that gastric cancer in this region may follow distinct pathogenetic mechanisms compared to those commonly seen in Asia [130].

Several factors underlie this disparity:**Strain Virulence**: East Asian *cagA* variants, characterized by ABD-type structures containing the EPIYA-D motif, and *VacA* with the s1/m1 genotype—both of which are more prevalent in Asia—are associated with increased oncogenic potential [83].**Gastritis Pattern**: Pangastritis predominates in East Asia and is associated with higher cancer risk, whereas Western patients more commonly exhibit antral-predominant gastritis [112,118,119].**Screening**: Endoscopic screening programs in Japan and Korea have led to earlier detection and improved outcomes [112,113,131]. Widespread screening may amplify the apparent incidence of gastric cancer beyond its true prevalence by detecting otherwise hidden cases.

It is unclear why gastric cancer incidence in Africa is low despite relatively high *H. pylori* prevalence (60–90%). This phenomenon, also named the African paradox, is probably a combination of bacterial, host, and environmental factors. This complicated issue is beyond the scope of this narrative review.

### 5.3. MALT Lymphoma

Mucosa-associated lymphoid tissue (MALT) lymphoma is a rare but clearly recognized consequence of *H. pylori* infection. It arises from chronic antigenic stimulation of B cells in the gastric mucosa [132]. Eradication therapy is a widely accepted first-line approach and can induce remission in many cases [133].

Although MALT lymphoma occurs worldwide, its detection may be more frequent in East Asia due to routine endoscopic surveillance [112,113,131]. Epidemiologic studies suggest higher relative prevalence in East Asian populations [134], while Western countries may underdiagnose early-stage cases due to lower prioritization of gastroscopy [135].

### 5.4. Gastritis and Histopathology

The pattern and severity of gastritis differ markedly between regions. Pangastritis, associated with more extensive mucosal damage and higher cancer risk, predominates in East Asia, whereas Western patients more commonly exhibit antral-predominant gastritis [112,118,119]. These differences are largely attributed to strain-specific virulence factors such as *cagA* and *vacA* s1/m1 alleles [83].

Asian patients often show more extensive atrophy, intestinal metaplasia, and lymphoid aggregates [136].

### 5.5. Extra-Gastric Manifestations

Beyond the stomach, *H. pylori* infection has been implicated in a growing number of extra-gastric conditions. Among the most well-documented are the following:**Iron-Deficiency Anemia (IDA)**: Multiple meta-analyses and clinical studies have confirmed a strong association between *H. pylori* infection and unexplained IDA, particularly in children and women of reproductive age. Eradication therapy has been shown to be able to improve hemoglobin levels in responsive cases [137,138].**Idiopathic Thrombocytopenic Purpura (ITP)**: The role of *H. pylori* in ITP is supported by numerous studies demonstrating platelet count recovery following eradication, suggesting an immunologic mechanism of action [137,138,139].**Vitamin B12 Deficiency**: Chronic *H. pylori*-induced gastritis can impair intrinsic factor production and parietal cell function, leading to malabsorption of vitamin B12. A prospective cohort study demonstrated that eradication of *H. pylori* alone significantly improved serum B12 levels and corrected anemia in affected individuals [137,138,140].**Metabolic and Cardiovascular Associations**: Although still a field with limited and evolving evidence, *H. pylori* infection has been linked to metabolic syndrome, insulin resistance, and cardiovascular disease through mechanisms such as low-grade systemic inflammation and molecular mimicry [138,141,142].

These findings underscore the systemic impact of *H. pylori* infection and support the broader consideration of eradication therapy beyond gastrointestinal indications [137,138,140,141]. *CagA*-positive *H. pylori* strains have been associated with increased risk of coronary artery disease and ischemic stroke through the progression of atherosclerosis [143,144,145]. While there is no definitive evidence demonstrating that the prevalence or severity of *H. pylori*-associated extra-gastric conditions varies according to regional strain differences, such variation may be possible.

### 5.6. Diagnostic Approaches and Treatment Strategies for H. pylori-Associated Diseases

Diagnostic and therapeutic approaches for *H. pylori*-associated diseases differ markedly between East Asian and Western countries, reflecting regional variation in disease burden, strain virulence, healthcare infrastructure, and clinical priorities.

In East Asia, where gastric ulcers and gastric cancer are more prevalent [109,129], endoscopic screening is widely implemented [112,113,131]. Endoscopic findings such as pangastritis, atrophy, intestinal metaplasia, and nodularity are routinely documented to stratify cancer risk [112]. Systems like the Kyoto classification [136] and the Kimura–Takemoto classification [146], both developed in Japan, provide standardized criteria for evaluating gastritis severity and predicting malignant potential based on endoscopic morphology.

Furthermore, magnifying endoscopy with narrow-band imaging (ME-NBI) has been widely adopted in Japan, allowing for enhanced visualization of microvascular and microsurface patterns and improving diagnostic accuracy for early gastric cancer [147,148,149]. These practices support early detection and enable timely therapeutic intervention.

Due to the high prevalence of gastric cancer, the need for minimally invasive treatment led to the development of endoscopic submucosal dissection (ESD), which offers excellent outcomes with organ preservation [150,151]. To guide post-ESD management, especially in cases of non-curative resection, the eCura scoring system was developed to stratify lymph node metastasis risk and determine the need for additional surgery [152]. In the surgical field, Japan has also contributed to the refinement of curative procedures, including the establishment of evidence-based lymphadenectomy strategies [153].

In Japan and South Korea, national screening programs [112,113,131] have contributed to low mortality rates despite high incidence [129] by facilitating early diagnosis and curative endoscopic treatment [147,148,149,150,151].

In contrast, although surgical procedures for gastric cancer—such as distal gastrectomy with Billroth I, Billroth II, or Roux-en-Y reconstruction—were originally developed in Western countries [154], progress in early detection and minimally invasive treatment has not outpaced that of East Asia over the past few decades [155]. Regarding diagnostic practices, Western countries rely more on symptom-driven endoscopy, largely due to the lower prevalence of gastric cancer. Considering healthcare resource allocation and cost-effectiveness, gastroscopy is less prioritized in routine practice [135,156]. However, in regions with relatively higher gastric cancer incidence, guidelines increasingly recommend *H. pylori* testing and endoscopic evaluation [156]. Awareness of early detection and the adoption of endoscopic therapy are gradually increasing in these areas, supported by updated recommendations from European and American societies [155,156].

## 6. Diagnostic Approaches

Differences in the diagnostic strategies exist, including in their use, availability, and accuracy. There are a number of different diagnostic strategies for the diagnosis of *H. pylori*. These diagnostic strategies can be grouped into invasive and non-invasive tests (Table 1). The main invasive tests require endoscopy and biopsy from the gastric mucosa. These include histopathology, rapid urease tests, culture, and molecular methods such as polymerase chain reaction (PCR) [157]. The diagnostic performance of these biopsy-based tests depends on the bacterial load in the samples, which may give false-negative results under low concentration; therefore, the biopsy specimens should be collected from a site with sufficient bacterial presence. It is recommended to obtain biopsies from both the gastric antrum and corpus, as collecting multiple samples from different mucosal sites increases sensitivity. Taking two biopsies—one each from the antrum and the corpus—is widely recommended [23,158,159]. To maximize test reliability, antibiotics and bismuth compounds should be discontinued for four weeks before testing, and PPIs should be discontinued for at least two weeks [160,161,162]. The accuracy of invasive approaches that require sampling from gastric mucosa is affected by conditions such as peptic ulcer bleeding, gastric atrophy, gastric carcinoma, and intestinal metaplasia, all of which can reduce the sensitivity of a test [158,163,164,165,166,167]. Non-invasive methods include the urea breath test (UBT), stool antigen test (SAT), antibody detection tests, and molecular methods [157,166,168,169]. Test selection and interpretation should be based on disease prevalence, potential for therapeutic interference, and cost [170]. Other things that should be taken into consideration are sensitivity and specificity, availability of a test, rapidity of the results, requirement for retrospective analysis or quantification, need for the detection of pathogenic properties, consistent reproducibility of the test, and the specific added value of the test [171].

It is universally accepted that no single test is considered the gold standard for infection diagnosis, and the reliability and accuracy of the diagnosis strengthen when multiple diagnostic tests are performed [23,36,172,173]. The performance of diagnostic methods varies as different gold standards are applied as a reference point. Diagnostic performance is also influenced by several factors, as previously mentioned. When diagnostic tests are performed correctly, most of them are highly accurate. Among 199 dyspeptic patients—who were defined as positive for infection when at least two test results were positive—showed that sensitivities and specificities were 95% and 100%, respectively, for the rapid urease test; 94% and 99%, respectively, for histopathological examination; 90% and 93%, respectively, for the stool test; and 90% and 90%, respectively, for the UBT [174]. Another study found that serum *H. pylori* IgG demonstrated a higher sensitivity (0.94) than urea breath and stool antigen tests (0.64 and 0.61, respectively) [170]. A comparative analysis of diagnostic techniques for *H. pylori* infection found sensitivity and specificity of histopathology (95/77%), rapid urease test (RUT) (100/84%), Gram staining (86/90%), IgG serology (100/67%), IgA serology (100/80%), PCR (100/75%), RUT and IgG serology combination (100/79%), and RUT, and Gram staining and IgG serology combination (100/92%), respectively. In that study, PCR emerged as the most reliable test with the highest accuracy of 94.2%. It was concluded that employing comparative detection methods rather than relying solely on one methodology is advisable for accurate detection [175].

### 6.1. Histopathology

Histological examination is one of the most commonly used diagnostic methods for *H. pylori* and is frequently regarded as the gold standard as it allows for direct detection of the pathogen. It has the advantage of simultaneous assessment of other types of gastric pathology such as gastritis, atrophy, intestinal metaplasia, and cancer [163,176,177,178,179,180,181,182]. It requires trained personnel for sample processing and interpretation, and is thus expensive. To minimize false-negative results, it is recommended that PPIs be discontinued for two weeks and antibiotics for four weeks before testing [158,163,166]. The presence of other bacterial species with structural similarity to Helicobacter can influence the results of this test [183,184]. Despite histology being the best available reference standard, there are occasional discrepancies between original histopathological reports and the reviews of biopsy slides. Consensus review by multiple pathologists provides the most reliable gold standard for diagnosis [165]. The diagnostic yield of histology is also influenced by the biopsy strategy. The Sydney system advises a minimum of five biopsies—one each from the lesser and greater curvature of the antrum, the lesser curvature of the corpus, the middle of the greater curvature, and the incisura angularis—to maximize detection [185,186]. Reduced sampling strategies have not consistently achieved diagnostic accuracy equivalent to the five-biopsy approach [187,188]. Routine hematoxylin–eosin (HE) staining is the most common technique, sometimes paired with Giemsa staining for more reliable detection of *H. pylori*. Although HE alone is sufficient in many cases, additional stains can be required when chronic active gastritis is observed without identifiable *H. pylori*, which may occur with low bacterial density or atypical localization [189,190]. Immunohistochemistry (IHC) provides the highest sensitivity and specificity and can give more diagnostic accuracy in patients on PPIs [191] but is not recommended as first-line treatment due to cost and time requirements [163,170,176,177,178,179,180,181,192]. Modified toluidine blue (MTB) has demonstrated superior performance compared to HE, particularly in post-eradication biopsies, small specimens with limited glands, or in the presence of excess mucus debris. In similar cases, other stains such as toluidine blue, acridine orange, Genta, Romanowski, or McMullen can be used for more accurate results [177]. Reported sensitivities for HE range between 70 and 95%, and can be improved with silver-based stains such as Genta or with IHC [193]. Some studies have found histology to have higher sensitivity and specificity than the UBT and the RUT for *H. pylori* diagnosis under special conditions, such as partial gastrectomy [194].

### 6.2. Culture

Culture of *H. pylori* from gastric biopsy specimens is not routinely performed in clinical practice. It is, however, considered a reliable diagnostic method under optimal conditions, with reported specificity approaching 100% and sensitivity around 70–90% [23,168,176,178,195,196]. The primary indication for culture is in cases where first-line eradication therapy has failed, as culture allows for evaluation of antibiotic susceptibility and subsequent selection of appropriate treatment [168,197,198,199]. The disadvantage of culture is that it is invasive, time-consuming, and demands resources such as skilled personnel and specialized laboratory facilities. The bacteria also require specific conditions such as reduced temperatures, microaerophilic environments, and enriched media, which are not available in all laboratories [158,200,201]. Factors that influence the accuracy of culture include the choice of culture medium, transport time and temperature, air exposure, the expertise of laboratory staff, and the bacterial load in biopsy samples [195,202,203,204]. False-negative results can arise if patients have taken PPIs within 2 weeks or antibiotics within 4 weeks prior to testing, similar to other diagnostic methods [205].

### 6.3. Rapid Urease Test

Urease is an enzyme produced by several bacterial species that hydrolyzes urea into ammonia and carbon dioxide [206]. Following biopsy collection, the specimen is placed in a solution containing urea and a pH indicator. The urease activity converts urea into ammonia and CO_2_, raising the pH and causing a visible color change in the indicator. Results may appear within minutes or may take several hours. It is important to consider how long to wait before interpreting a positive result, as different types of RUT show varying sensitivity depending on the time elapsed before a test turns positive. Commercially available RUT kits recommend making a decision (positive vs. negative) within 24 h. The time it takes for a test to show a positive result depends on both the bacterial concentration and the surrounding temperature. Most tests will become positive within 120 to 180 min, but it is advisable to retain any that initially appear negative for the full 24 h period. Laine L, et al. reported that interpreting the rapid urease test within three hours is associated with a false-negative rate of approximately 40% [207]. One study aimed to evaluate both the diagnostic accuracy and reaction time of Pronto Dry versus a liquid-phase rapid urease test, both before and after treatment of *H. pylori* infection. In untreated patients, the sensitivity of Pronto Dry at 5, 15, 30 min, and 3 and 24 h in untreated patients was 45%, 71.2%, 81.1%, 90.1%, and 91.9%, respectively, for the Pronto Dry vs. 6.3%, 31.5%, 51.3%, 78.4%, and 90.1% for the liquid-phase rapid urease test [208]. In a study evaluating three different commercially available RUTs, researchers found that Pyloritek demonstrated significantly shorter mean and median times to a positive result (0.5 ± 0.02 h and 0.5 h, respectively) compared to CLOtest (2.0 ± 0.6 h and 0.75 h) and Hpfast (2.2 ± 0.6 h and 0.5 h). The study concluded that when a result is needed within one hour, Pyloritek offers higher sensitivity than either CLOtest or Hpfast, without compromising specificity [209]. In the study by Yousfi MM, et al., the sensitivity of the CLOtest continued to increase up to 24 h post-biopsy, while no false-positive results were observed during this period [210], although some studies require careful interpretation; positive results after 24 h are most often false positives and should be carefully interpreted [211]. The sensitivity and specificity values of this method across most studies are more than 90% [176,212]. In cases where the RUT result is negative, a confirmation is needed using appropriate alternate tests. Several non-*H. pylori* organisms, including *Klebsiella pneumoniae*, *Staphylococcus aureus*, *Proteus mirabilis*, *Enterobacter cloacae*, and *Citrobacter freundii*, also produce urease and may cause false-positive results when present in the stomach or oral cavity [213,214,215].

### 6.4. Urea Breath Test

The urea breath test (UBT) has higher diagnostic accuracy than other non-invasive tests to identify *H. pylori* (in patients without a history of gastrectomy) [168,216]. It evaluates the conversion of urea using isotopically labeled carbon. After oral administration of urea labeled with either ^13^C or ^14^C, the urease enzyme produced by *H. pylori* hydrolyzes the compound into ammonia and carbon dioxide. ^13^C-labeled urea is preferred over ^14^C since it is stable and non-radioactive. The carbon dioxide generated by this process diffuses across the gastric epithelium, enters the bloodstream, and is ultimately exhaled. Detection of isotope-labeled CO_2_ in breath samples, typically within 10 min, serves as an indicator of *H. pylori* infection [158,217,218].

The urea breath test offers several advantages as it is non-invasive, safe, and accurate [219,220]. Among patients without a history of gastrectomy and who have not recently used antibiotics or PPIs, UBT demonstrates superior diagnostic accuracy compared with serology and stool antigen tests [217]. Variability in test results is mainly influenced by the administered dose of ^13^C-labeled urea and the method of respiratory sample collection. Adjusting these aspects of the testing protocol could further enhance accuracy [221]. Moreover, in regions where access to endoscopy is limited, UBT can serve as a reliable diagnostic alternative [222]. Conventional UBT involves the use of ^13^C-labeled urea diluted in orange juice. Results can be affected by urease-producing oral flora, leading to increased ^13^CO_2_ excretion in earlier samples and potential false-positive results. A capsule-based ^13^C-labeled urea test, administered at a lower dose than the conventional method, has shown high sensitivity and specificity [223]. Results from film-coated ^13^C-labeled urea tablets may still be influenced by non-*H. pylori* urease-positive bacteria. Evidence suggests that such bacteria within the stomach, rather than in the oral cavity, may cause false positives as UBT values measured at 20 min are significantly higher than those at 5 min. In cases of reduced gastric acidity, translocation of urease-positive bacteria from the oral cavity to the gastric mucosa may also contribute to false-positive outcomes [213].

False-negative results can arise if patients have taken PPIs within 2 weeks or antibiotics within 4 weeks prior to testing [224]. Discontinuation of PPIs prior to testing not only restores *H. pylori* bacterial load but also lowers gastric pH, which increases ^13^CO_2_ production since the urease enzyme is most active in acidic environments [225,226]. Gastrointestinal bleeding also reduces the diagnostic accuracy of UBT, and testing is therefore recommended after bleeding has resolved [227]. Additionally, corpus-predominant gastritis may yield false-negative outcomes, as advanced atrophic changes reduce gastric acid secretion and elevate intragastric pH [228]. In contrast, corpus-predominant gastritis has also been reported as a potential cause of false-positive results in UBT as autoimmune gastritis may promote the overgrowth of non-*H. pylori* urease-producing bacteria [229].

### 6.5. Stool Antigen Tests

The stool antigen tests (SATs) detect *H. pylori* antigens excreted in the feces of infected individuals [230]. Two main types of SATs are available, enzyme immunoassay (ELISA) and immunochromatography assay (ICA), both of which may use either polyclonal or monoclonal antibodies [168]. The ELISA-based SAT is considered a good alternative when ^13^C-UBT is unavailable, provided that monoclonal antibodies are used as reagents, since, compared with a polyclonal enzyme immunoassay, the monoclonal stool antigen test has shown better sensitivity and specificity [231,232]. ICA, on the other hand, offers the advantage of rapid diagnosis and is particularly useful in developing countries or remote areas with limited access to healthcare. It is also easy to store in many hospitals, and can be performed in small laboratories without the need for specialized equipment or personnel [233]. When properly conducted, the sensitivity and specificity of SAT have been reported as 94% and 97%, respectively [37]. A meta-analysis evaluating SATs for diagnosis in children found similar diagnostic accuracy before and after treatment, with sensitivity and specificity remaining consistent [234]. The SAT does not require fasting and, moreover, newer monoclonal antibody-based assays have been developed that remain unaffected by PPI use, further improving the reliability of the test [168,235]. These advantages can make the SAT a better alternative to UBT [37]. However, like UBT, SAT accuracy can be affected by factors that reduce bacterial load, such as gastrointestinal disorders, PPIs, antibiotics, N-acetylcysteine (NAC), bismuth compounds, and bleeding ulcers, all of which can lead to false-negative results [235,236]. Further limitations of the SAT test include patient reluctance to handle stool samples and improper storage, which can compromise results. To preserve antigen integrity, samples should be frozen if not tested promptly, which may be problematic in regions without access to freezing facilities. Additionally, the choice of cutoff values affects test sensitivity and specificity, which may vary across populations; therefore, local validation is needed to optimize accuracy [163]. As well as UBT, the SAT has the advantage of distinguishing between active infection and past infection, which is why it is useful for monitoring eradication therapy [23].

### 6.6. Antibody Detection Tests

Serological tests for *H. pylori* are primarily based on the detection of antibodies against the bacteria. IgG, IgA, and IgM can be measured, but IgG provides the most reliable results. IgM and IgA have limited clinical value due to their lack of precision and higher costs, especially when more accurate alternatives exist [222,237,238]. Antibody detection is commonly performed using immunoblotting or enzyme immunoassays, including ELISA with serum, saliva, or urine samples. Serum-based ELISA is the most widely applied antibody test for *H. pylori* [168,217,239]. Antibody detection using samples from urine and saliva has also demonstrated high sensitivity and specificity. These samples are inexpensive and easy to collect without specialized skills or equipment [222]. However, antibody levels in urine and saliva are lower than in serum, which may reduce diagnostic accuracy [164,222]. In addition to conventional assays, multiplex serology enables detection of antibodies to multiple protein antigens [239]. Studies have applied multiplex serology with twelve *H. pylori* antigens [240] or fifteen proteins from three different strains [239,241]. Multiplex serology may also support risk stratification and targeted prevention strategies when integrated with other predictive factors [242]. Reported sensitivity and specificity for this method range between 76 and84% and 79 and90%, respectively [176]. In patients receiving colloidal bismuth, antibiotics, or PPIs, IgG-based serology can be useful because it is less affected by suppression of *H. pylori* [11]. Accordingly, serology remains efficient as an initial diagnostic tool in cases where there is gastrointestinal bleeding, gastric carcinoma, MALT lymphoma, and atrophic gastritis since the test’s accuracy is not compromised by these conditions, which may commonly cause false negatives in other diagnostic methods [158,231]. It also provides advantages of low cost, broad availability, and simplicity [198,243,244]. The test has also proven useful for diagnosing *H. pylori* infection in children [244]. It is useful as an exclusion test, owing to its high sensitivity and negative predictive value [170]. However, in regions with low *H. pylori* prevalence, serology is not recommended, as the probability of false-positive results increases. This suggests that the positive predictive value of the test decreases with decreasing prevalence of *H. pylori* [245]. Since the performance of these tests varies geographically, depending on the antigenic composition of the circulating strains, age, sex, and ethnicity, careful kit selection is important, and locally validated assays are essential [11,170]. Validated antibody detection kits are valuable for initial screening before the diagnosis is confirmed by histology or culture in areas with limited access to endoscopy [222,236]. Using this method in dyspeptic patients can reduce healthcare costs and unnecessary endoscopies [222]. The major disadvantage of serology is its inability to differentiate between active infection and prior exposure [164,170,222]. Following eradication therapy, antibodies may persist for several months, and so it cannot be used to confirm eradication [163,222,243,246]. After eradication of the bacterium, IgG decreases slowly and only after approximately 6 months, a decrease of 50% in the antibody titer is observed [247,248]. Furthermore, IgG antibody levels are known to remain relatively high, even for several years following eradication therapy. Tanaka, et al. reported that a significant reduction in *H. pylori* antibody titer occurs within 1 year after eradication treatment, but a long period is needed to achieve complete negative conversion [249]. This persistence often leads to false positives, even after bacterial clearance [170]. If UBT or SAT is available, serology is not preferred for initial diagnosis since it only indicates past exposure [163]. IgG antibodies usually appear about 21 days after infection so false negatives can also occur in early infection, when antibody titers are not yet sufficiently elevated [245].

### 6.7. Molecular Methods

Over the past few decades, molecular detection has significantly improved the clinical management of many infectious diseases [250]. Among molecular methods, PCR is considered one of the most effective, with wide clinical applications [251,252]. PCR-based detection of *H. pylori* can be performed using both invasive and non-invasive approaches [250]. Commonly analyzed samples include gastric juice, biopsy tissue, saliva, and feces [251,253,254]. PCR is highly sensitive and specific (>95%) [38,253,255]. It provides a simple, rapid, and accurate method for detecting *H. pylori* [251,254]. Compared to other diagnostic techniques, PCR demonstrates superior accuracy in patients with gastrointestinal bleeding [38,253]. Successful PCR requires careful primer design and appropriate gene selection [256]. Target genes such as *vacA*, *cagA*, *ureA*, *glmM*, *HSP60*, *16S rRNA*, *23S rRNA*, and *ureC* are commonly amplified to detect the *H. pylori* genome [176,257]. Using two or more target genes further improves diagnostic specificity and reduces false positives, particularly in non-biopsy specimens [38,253,254]. PCR has also been applied in environmental studies to identify *H. pylori*. Its presence has been confirmed in drinking water [258], and higher detection rates in unwashed vegetables suggest that proper washing of produce may help reduce infection risk [259]. However, PCR has limitations, including high cost, the need for skilled personnel, and potential false positives due to amplification of DNA fragments from non-viable bacteria [251,252,260].

The rise in bacterial resistance to antibiotics has become a growing challenge for healthcare [261]. Molecular approaches such as PCR are valuable not only for pathogen detection but also for identifying resistance mutations, thereby supporting appropriate treatment strategies [262]. To determine *H. pylori* antibiotic susceptibility, culture from gastric biopsy still remains the gold standard; however, culturing from gastric biopsy is invasive, skill-dependent, and time-consuming, and new methods have been developed [263]. Molecular genetic assays, including qPCR, are being increasingly utilized for the identification of *23S rRNA* and *gyrA* mutations, which confer resistance to clarithromycin and levofloxacin, respectively [264,265,266]. A high concordance has been observed between these mutations and the corresponding antibiotic susceptibility phenotype [267]. qPCR is, thus, a fast and highly sensitive method for assessing *H. pylori* antibiotic resistance. qPCR can also be employed for the direct detection of *H. pylori* and has demonstrated an overall concordance rate of 95.9% with the results of the ^13^C-labeled urea breath test (^13^C-UBT) [268]. Another study found the sensitivity and specificity of qPCR in detecting *H. pylori* to be 96.4% and 88.5%, respectively [266]. The string test (also called the Entero-test) is a quick and minimally invasive test to obtain gastric samples, and involves swallowing an encapsulated string that is then retrieved orally, and allows for further studies on the accompanying gastric material—such as the detection of *H. pylori* with culture or PCR [269,270,271]. An innovative way of susceptibility-guided therapy includes the use of qPCR on samples collected from the string test. Han et al. demonstrated a high treatment success when using the string–qPCR method, indicating that the method is a valuable, non-invasive, rapid, and accurate way to improve *H. pylori* treatment success [268].

### 6.8. Post-Treatment Testing

For post-treatment testing, most major clinical practice guidelines recommend confirming eradication using either the urea breath test or stool antigen test [23,166,272,273]. Confirming eradication is critical as treatment failure may occur, and persistent infection increases the risk of complications such as peptic ulcer disease and gastric cancer. Confirming eradication also provides useful insight into antibiotic-resistance patterns [272,273]. No guideline explicitly favors one test over the other, and the choice of method should be determined in the context of geographical location, availability of the test, alongside factors that influence each test’s accuracy. SATs may be more appropriate in regions where the *H. pylori* prevalence exceeds 30% [274], whereas in lower-prevalence areas, results may be best verified with UBTs [274,275]. Both non-invasive methods, however, are influenced by fragments of bacterial cells, the recent use of antibiotics, bismuth compounds, and acid-suppressing drugs, which must be discontinued before testing. For this reason, eradication testing is generally performed 4 to 6 weeks after treatment, allowing sufficient time for the residual bacterial component to be cleared and the effects of these agents to wear off [23].

### 6.9. East vs. West Diagnostic Strategies

As the focus of this review is to compare differences between the West and East, we sought to determine whether there are differences in diagnostic strategies between regions.

According to the American College of Gastroenterology clinical guidelines from 2024 [23], indications for testing and treatment of *H. pylori* are as follows: (i) past or present peptic ulcer disease, (ii) gastric MALT lymphoma, (iii) uninvestigated dyspepsia in patients who are under the age of 60 (in high-risk populations for gastric cancer, test and treat at age 45–50 years), (iv) functional dyspepsia, (v) adult household members of individuals who have a positive non-serological test for *H. pylori*, (vi) patients taking long-term NSAIDs or starting long-term treatment with low-dose aspirin, (vii) patients with unexplained iron-deficiency anemia, (viii) patients with idiopathic thrombocytopenic purpura, (ix) as a primary and secondary prevention of gastric adenocarcinoma (current or prior history of gastric premalignant conditions, current or prior history of early gastric cancer resection, current or prior history of gastric adenocarcinoma, patients with gastric adenomas of hyperplastic polyps, persons with a first-degree relative with gastric cancer, individuals at increased risk for gastric cancer including certain non-White racial/ethnic groups, immigrants from high gastric cancer incidence regions/countries, hereditary cancer syndromes associated with an increased risk for gastric cancer, and patients with autoimmune gastritis). The indications for diagnosis and treatment are mostly the same in major guidelines or consensus report, such as the *Maastricht VI/Florence Consensus Report* [276], the *Fifth Chinese National Consensus Report On the Management of Helicobacter Pylori Infection* [277], the Kyoto Global Consensus Report [272], the Taipei Global Consensus [278], and Korean guidelines [279]. There is no universal consensus on the most appropriate screening test, as the decision depends on whether invasive or non-invasive approaches are suitable. Clinical guidelines emphasize the importance of selecting and performing appropriate diagnostic tests at the optimal timing, based on regional epidemiology, population characteristics, and individual patient risk profiles [23,272,273,276,277,278,280].

***Diagnostic Approach***: The use of non-invasive testing (e.g., UBT) and the “test-and-treat” strategy is recommended according to American and European guidelines, with regional epidemiology influencing its application. These strategies are recommended for younger patients with no alarming symptoms. It is widely accepted that endoscopy should be reserved for patients with symptom onset after 50 years of age, those who have alarm features, and all patients who fail empirical antisecretory therapy or the test-and-treat strategy [23,276]. In contrast, in East Asia, this strategy is often integrated with gastric cancer screening programs. In Japan and other Asian countries, ABC screening—which combines serum antibody and pepsinogen level—is performed as a primary risk stratification for gastric cancer [281]. It is agreed, according to the Taipei Global Consensus for screening and eradication of *H. pylori* for gastric cancer prevention, that the strategy of screen-and-treat for *H. pylori* infection is most cost-effective in young adults for gastric cancer prevention in regions with a high incidence of gastric cancer like in Eastern Asia. The Taipei Consensus agreed that population-wide screening and eradication of *H. pylori* infection should be integrated or included in the national healthcare priorities to optimize resources [278]. A large-scale clinical trial (HELER), currently being conducted in the Republic of Korea, is exploring the effectiveness of *H. pylori* eradication as a primary prevention method for the general population. The study aims to provide robust evidence on the potential of *H. pylori* eradication in preventing gastric cancer among individuals at average risk [279].

***Role and Timing of Endoscopy****:* National screening programs using endoscopy for gastric cancer are implemented only in Japan and Korea, with Mongolia joining in 2022 [112,113,131,282]. In contrast, Western guidelines do not recommend endoscopic screening due to low prevalence and cost-effectiveness concerns; however, among higher-risk subgroups, such as older patients and migrants from high-prevalence areas, endoscopy should be conducted [23,276].

***Purpose of Eradication****:* In Asian countries, *H. pylori* eradication is clearly positioned as a strategy for gastric cancer prevention. In Western countries, the primary goals are ulcer healing and symptom relief, with cancer prevention as a secondary consideration [23,272,276,277,278,279].

***Requirement for Eradication Confirmation****:* Guidelines vary in whether post-treatment confirmation is mandatory. According to American and European guidelines, all patients who are treated for *H. pylori* infection should undergo a test of cure at least 4 weeks after completion of therapy [23,276]. According to the Kyoto Consensus Report, the outcome of eradication therapy should always be assessed, preferably non-invasively [272]. The Taipei Global Consensus for screening and eradication of *H. pylori* for gastric cancer prevention agreed that a confirmation test of *H. pylori* eradication is not mandatory in mass screening, but should be performed in subsets of the population for assessment of treatment efficacy [278].

## 7. Antibiotic Resistance and Treatment Strategies

After the diagnosis of *H. pylori*, the most important and obvious step in the management of this chronic infection is eradication of the bacteria. Previously, the indication for eradication of *H. pylori* was based on certain defined conditions, but according to recent guidelines, all infected patients should be treated, with or without gastrointestinal symptoms or lesions [11,276].

Left untreated, *H. pylori* infection can lead to serious consequences such as relapse of duodenal and gastric ulcers, peptic strictures, gastric adenocarcinoma, and mucosa-associated lymphoid tissue lymphoma. Thus, highly effective eradication therapy is of major importance. According to international guidelines, the first-line treatment in countries with a low clarithromycin resistance (<15%), traditional triple therapy, should be undertaken with a high-dose proton pump inhibitor (PPI), clarithromycin, and amoxicillin or metronidazole [11,276]. Apart from clarithromycin-based therapies, Bismuth-containing quadruple therapy (BQT) with bismuth salicylate, PPI, tetracycline, and metronidazole has also been suggested as an option in uncomplicated cases [11,276,283].

Due to an increased antibiotic resistance of *H. pylori* worldwide, therapy recommendations have to take into consideration the local resistance patterns. In this current narrative review, we were not able to obtain reliable information on whether the antibiotic resistance was primary or secondary, which is a limitation.

In general, it has been recommended that in areas of high clarithromycin resistance (>15%), therapy should be selected based on the prevalence of metronidazole resistance and dual metronidazole–clarithromycin resistance. Where metronidazole resistance is low, treatment containing PPI, amoxicillin, and metronidazole is recommended, whereas in areas of low dual metronidazole–clarithromycin resistance, BQT or concomitant non-bismuth-containing quadruple therapy is recommended [11,276,283]. However, BQT has been recommended in countries and areas where dual clarithromycin–metronidazole resistance is >15% [11,276,283]. Unfortunately, bismuth-based treatment is not available in many countries. According to a recent international survey, more than 1 billion inhabitants worldwide do not have access to bismuth-based regimens (287). In countries where BQT is not available, other combinations of drugs are obviously needed, and other factors need to be taken into consideration, such as number of different antibiotics, duration of therapy, dose, or type of PPIs. A longer duration of the eradication therapy can, in some instances, improve the eradication rate [283].

Interestingly, the new potassium-competitive acid inhibitor, vonoprazan (VPZ), at 20 mg per day has been shown to be more effective than lansoprazole (30 mg per day) in patients treated with standard clarithromycin-based triple therapy (STT) (93% vs. 76%), as well as in patients with clarithromycin resistance (82% vs. 40%) [284]. Other studies from Japan have also shown good results with vonoprazan in terms of eradication success [285,286,287]. Jiang et al. demonstrated 94% and 98% eradication rates for the combination of amoxicillin 1 g × 3 and vonoprazan 20 mg 1 × 2 for 2 weeks in intention treatment and per-protocol analysis [288]. In a provocative paper, Waldum and Fossmark suggested that perhaps vonoprazan alone could be an option in the future for eradication of *H. pylori* [289].

### 7.1. Antibiotic Resistance in the West and the East

*H. pylori* resistance to antibiotics has been shown to be very high in some areas in the world [290]. This is, however, quite different for different antibiotics in different countries and regions in the world [291]. For example, in Egypt, *H. pylori* resistance for clarithromycin, levofloxacin, metronidazole, amoxicillin, and tetracyclines was 40–53% (C), 20% (L), 100% (M), 82–95% (A), and 25–38% (T) [292,293]. However, in Iceland, much lower antibiotic-resistance patterns were observed: 9% (C), 4% (L), 1% (M), 0% (A), and 0% (T) [294]. In general, clarithromycin resistance is higher in China and Japan, around 35% higher than in Europe and North America (Table 2), whereas resistance was similar in Korea and the West (Table 1). Levofloxacin has a strong bactericidal effect against *H. pylori*, and in regions with high clarithromycin resistance, levofloxacin has been suggested as first-line and second-line eradication treatment, particularly where bismuth therapy is not available [276]. However, in Japan, levofloxacin is not used in the eradication of *H. pylori* and sitafloxacin is recommended [295]. No pattern of resistance seems to be present between the West and the East in terms of metronidazole, with a high resistance in China, around 80% [296] (Table 1), whereas this is very low at only 4% in Japan [297]. However, in Korea [298], metronidazole resistance is around 30%, similar to rates in Europe [291] and in North America [299]. A limitation of the review of antibiotic resistance for Europe is that very limited data were available from Eastern Europe, and the data from Europe are only from the Western part of Europe.

### 7.2. Guidelines for H. pylori Eradication in the West and the East

#### 7.2.1. General Aspects

Different strategies have been proposed to select patients with upper abdominal symptoms for a gastroscopy. According to recent guidelines on the management of dyspepsia, patients ≥60 years of age presenting with new onset dyspepsia should be investigated with upper endoscopy to exclude clinically relevant findings such as gastrointestinal malignancy [300]. In recent European guidelines, it is stated that, “in younger patients without alarming symptoms”, there is no need to perform a gastroscopy to detect malignancy, which is “rare” [301].

International guidelines recommend a non-invasive “test-and-treat” strategy for *H. pylori* in the management of uninvestigated dyspepsia in patients < 45–55 years without red flags (alarm features), if the local *H. pylori* prevalence exceeds 10–20%, which is the most cost-effective [11,276,302]. Cure rates of susceptibility-guided vs. empirical treatment of *H. pylori* have been compared but results are controversial [303,304].

#### 7.2.2. Eradication Therapy in the West

In Europe and North America, in countries with a relatively low clarithromycin resistance (<15%), classical triple therapy should be undertaken with a high-dose proton pump inhibitor (PPI), clarithromycin, and amoxicillin or metronidazole [11,276,283]. For example, in Northern Europe, the clarithromycin resistance is lower than 10% [294,305,306,307,308]. In other parts of Europe, the resistance patterns are lower by 12–22% and in North America by 17–19% than in China and Japan (34–36%), as shown in Table 1. However, in patients infected with *H. pylori* with clarithromycin resistance, the cure rates with standard triple therapy are inadequate, with only around 40–75% cured [309].

Non-bismuth quadruple treatment has been shown to have a much better efficacy in comparison with sequential therapy in studies with clarithromycin-resistant *H. pylori* strains [276]. This regime has also demonstrated adequate efficacy in metronidazole-resistant, clarithromycin-susceptible cases because of its PPI–amoxicillin–clarithromycin component. In the International multicenter prospective European Registry on *Helicobacter pylori* Management (Hp-EuReg), approximately 30,000 patients from 27 European countries were evaluated, and 78% first-line empirical *H. pylori* treatments were included for analysis [310,311]. Overall, 23% of strains were resistant to clarithromycin, 32% to metronidazole, and 13% to both. Standard triple therapy with amoxicillin and clarithromycin was the most common treatment strategy (39%), achieving an 82% intention-to-treat eradication rate. However, >90% eradication was obtained only with 10-day bismuth quadruple or 14-day concomitant treatments. Adequate compliance, higher levels of acid inhibition, and longer therapy durations were associated with higher eradication rates. As has been pointed out, patients are exposed to at least one unnecessary antibiotic with quadruple therapy, potentially increasing the risk for antimicrobial resistance globally [276,312].

#### 7.2.3. Eradication in the East: China, Japan, and Korea

Treatment guidelines for eradication therapy for *H. pylori* in China, Japan, and Korea have been revised recently [313]. Antibiotic resistance is somewhat different in these Asian countries (Table 1), which makes *H. pylori* eradication strategies different.

In China, 14 days of treatment is recommended with bismuth quadruple, with different combinations of PPIs (bid), bismuth, clarithromycin, metronidazole, levofloxacin, and furazolidone (nitrofurantoin antibiotic). However, the recommended *H. pylori* therapies were not categorized into first- and second-line regimens [313].

In Japan, a 7-day course is recommended. The first-line therapy for eradication of *H. pylori* consists of a PPI or vonoprazan 20 mg once daily, amoxicillin 750 mg bid, and clarithromycin 200 mg bid. The second-line therapy should be a combination of PPI bid, amoxicillin 750 mg, and metronidazole 250 mg bid. The third-line therapy consists of PPI bid, amoxicillin 750 mg, or metronidazole 250 mg bid, as well as sitafloxacin 100 mg bid [295,314,315].

In Korea, the first standard therapy, as practiced in the West, consists of a PPI bid, amoxicillin 1000 mg bid, and clarithromycin 500 mg bid, although the duration of therapy is 14 days. The much lower clarithromycin resistance in Korea than in China and Japan is probably the basis for this regime (Table 1). Sequential therapy is then recommended for 10 days: PPI bid + AMX 1000 mg bid for 5 days, then clarithromycin 500 mg bid and metronidazole 500 mg bid for 5 days. First-line tailored therapy is recommended for 7–14 days: standard triple therapy (CAM-sensitive) or classic BQT (in clarithromycin-resistant strains). Second-line therapy consists of a duration of 10–14 days, classic BQT, i.e., PPI bid and bismuth 120 mg qid, metronidazole 500 mg tid, and tetracycline 500 mg qid. Finally, third-line treatment is recommended as PPI bid, amoxicillin 1000 mg bid, and LVFX 500 mg qd or 250 mg bid [316,317,318].

In summary, eradication strategies for *H. pylori*, are quite different in these large countries in the East—China, Japan, and Korea. In China, bismuth is always included in the recommended combination therapy and therapy is undertaken for 14 days. In Japan, it seems that vonoprazan is taking over as an acid-inhibitory therapy and is associated with better eradication rates. In Korea, standard triple therapy is commonly used due to the lower clarithromycin resistance, and second-line and third-line treatments with levofloxacin are the standard of care.

## 8. Conclusions

This review highlights the geographic divergence of *H. pylori* in terms of epidemiology, strain diversity, clinical burden, and management strategies. Eastern countries, especially East Asia, face a higher disease risk, particularly for gastric cancer, due to more virulent strains and distinct patterns of gastritis. In contrast, Western countries contend with lower prevalence but variable resistance profiles and screening uptake. By comparing six key domains—prevalence, transmission, genetic structure, clinical outcomes, diagnostics, and treatment—this review underscores the need for region-specific approaches to screening, prevention, and eradication, while also reflecting recent advances in understanding of *H. pylori* biology and regional variation. Continued genomic surveillance and international collaboration will be essential to reduce disparities and improve global outcomes.

## Figures and Tables

**Figure 1 ijms-26-11408-f001:**
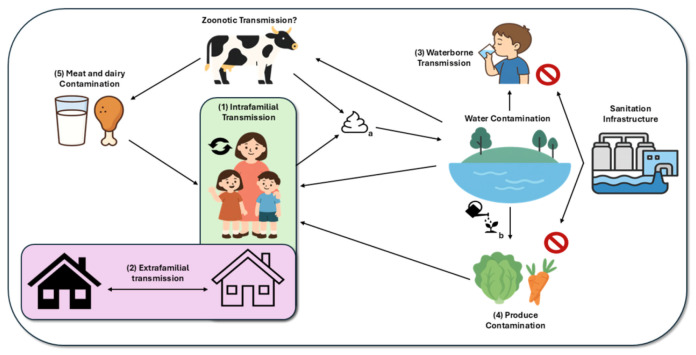
A summary of *H. pylori* transmission. (**1**) Intrafamilial transmission: likely occurring via the gastro-oral or fecal–oral route. (**2**) Extrafamilial transmission: more common in rural and developing settings. (**3**) Waterborne transmission. (**4**) Produce contamination: arising from irrigation with contaminated water. (**5**) Meat and dairy contamination: a potential zoonotic source of foodborne transmission through exposure to infected livestock. (**a**) Pathway of water contamination in soil via human or animal feces. (**b**) Pathway of contamination of produce via soil water.

**Table 1 ijms-26-11408-t001:** Diagnostic methods categorized by invasiveness, principles, sample types, advantages, and disadvantages.

Diagnostic Method	Invasiveness	Principle	Sample Type	Advantages	Disadvantages/Limitations
**Histopathology**	Invasive	Microscopic detection of *H. pylori* and gastric pathology	Gastric biopsy	Enables detailed assessment of gastric mucosa	Requires biopsy and expert interpretation; expensive; false negatives with low bacterial load
**Culture**	Invasive	Growth of *H. pylori* on selective media	Gastric biopsy	Allows for antibiotic susceptibility testing and strain typing	Technically demanding; slow; requires specialized lab and fresh samples; false negatives with low bacterial load; false negatives with recent PPI/antibiotic use
**Rapid Urease Test (RUT)**	Invasive	Detection of urease enzyme activity	Gastric biopsy	Simple, rapid, inexpensive	False positives from other urease-producing bacteria; false negatives with recent PPI/antibiotic use
**Urea Breath Test (UBT)**	Non-invasive	Urease-mediated hydrolysis of labeled urea (^13^C or ^14^C)	Exhaled breath	Non-invasive, useful for both diagnosis and post-treatment assessment	Affected by recent antibiotic or PPI use and GI bleeding
**Stool Antigen Test (SAT)**	Non-invasive	Detection of *H. pylori* antigens in stools	Stool	Simple, non-invasive, useful for post-treatment assessment; unaffected by PPIs (newer monoclonal assays)	Requires proper sample storage; patient reluctance; affected by PPIs, antibiotics, GI bleeding
**Serology (IgG/IgA)**	Non-invasive	Detection of antibodies against *H. pylori* in serum	Serum, saliva, urine	Inexpensive; widely available; unaffected by gastric bleeding or PPI use; can support risk stratification (multiplex serology)	Cannot distinguish active from past infection; false positives post-eradication
**Molecular Methods**	Invasive/Non-invasive	Amplification of target genes, detection of mutations (e.g., *vacA*, *cagA*, *ureA*, *23S rRNA*)	Biopsy, saliva, stool, gastric juice	Sometimes non-invasive, highly sensitive and specific, allows for susceptibility testing, strain typing, and detects resistance mutations	Expensive; need for skilled personnel; risk of false positives from non-viable DNA

**Table 2 ijms-26-11408-t002:** Antibiotic resistance for *H. pylori* in the West (Western Europe and North America, USA, and Canada), and the East (China, Japan, and Korea), as well as the extremes of the spectrum found in Egypt, with very high antibiotic resistance, and Iceland, with low antibiotic resistance.

	Clarithromycin	Levofloxacin	Metronidazole	Amoxicillin	Tetracycline
Western Europe (Germany, France, and UK)	12–22%	13–23%	17–62%	0–3.5%	0–0.5%
North America (USA, and Canada)	17–19%	43%	29–35%	1%	2%
-China	34%	35%	78%	3%	2%
-Japan	36%	lack of data	4%	3%	lack of data
-Korea	18%	37%	30%	10%	0%
Egypt	40–52%	20%	100%	82–95%	17–42%
Iceland	9%	4%	1%	0%	0%

## Data Availability

No new data were created or analyzed in this study. Data sharing is not applicable to this article.

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
