# Peer review of "Helicobacter pylori* Across Continents: Contrasts in Epidemiology, Genetics, Clinical Impact, and Management Between East and West"

_ijms, 2025, doi:10.3390/ijms262311408_

Round 1
Reviewer 1 Report
Comments and Suggestions for Authors
Introduction
Please define “East” and “West”: regions, countries.
Please mention the type of review (narrative).
Please emphasize the importance and the novelty of this review.
2. Epidemiology and prevalence
Please foremost specify data from those countries considered as belonging to East, respectively West, since presenting data from all over the world deters attention from the main intended comparison of this review.
6.2 Culture
Please add: “false negative results can arise if patients have taken PPIs within 2 weeks or antibiotics within 4 weeks prior testing” – as stated in line 61 of subsection 6.4 Urea breath test.
Please verify the number of references, ex. 232 in the text corresponds to 234 from the reference list, 233 in the text corresponds to 235 from the reference list, 239 in the text corresponds to 241 from the reference list ….
7.1 Antibiotic resistance
When referring to antibiotic resistance, please specify in every instance whether primary or secondary resistance is meant.
Table 1. There are significant differences between Eastern and Western European countries; perhaps it would be a good idea to refer only to Western European countries, or, even better, to separate the first line of the table into Western Europe and Eastern Europe (studies from Eastern Europe should also be identified and included, for a comprehensive review).
Conclusions – please provide a distinct conclusions paragraph, summarizing the main findings of this review.
Author Response
Introduction, Please define “East” and “West”: regions, countries.
Response: Thank you very much for pointing this out. We have now clarified the geographic scope of “East” and “West” in the context of this review. Specifically, “East” refers to Asia and the Middle East, with particular emphasis on East Asia (e.g., Japan, China, South Korea). “West” refers to Europe and the Americas, with a focus on Western Europe (e.g., Germany, France, the United Kingdom) and North America (e.g., the United States and Canada). This distinction is now explicitly stated in the revised Introduction (lines xx–xx).
To avoid confusion prior to this definition, we have also replaced earlier instances of “the West” with “Western countries.” Thank you again for helping us improve clarity and consistency.
Please mention the type of review (narrative).
Response: Thank you for this helpful suggestion. We have now specified that this is a narrative review, which synthesizes recent findings without conducting a systematic meta-analysis. This clarification has been added to the Introduction (line 64).
Please emphasize the importance and the novelty of this review.
Response: Thank you very much for this important suggestion. In response, we have revised the final paragraph of the Introduction to better highlight both the significance and the originality of this review. Specifically, we emphasize that the strength of this narrative review lies in its comprehensive synthesis of the most recent findings on H. pylori, as well as its focused comparison between Eastern and Western regions. By integrating up-to-date data and examining geographic disparities in epidemiology, strain virulence, diagnostic practices, and treatment strategies, the review offers timely insights that are highly relevant to current clinical and public health challenges. We believe this dual emphasis—on recency and regional comparison—adds meaningful value to the literature and supports the need for more tailored approaches to H. pylori management worldwide.
2. Epidemiology and prevalence
Please foremost specify data from those countries considered as belonging to East, respectively West, since presenting data from all over the world deters attention from the main intended comparison of this review.
Response: Thank you very much for this valuable observation. In response, we have restructured Section 2 to follow a geographically comparative framework, beginning with prevalence data from countries broadly categorized as “East” and “West” within the scope of this review. This prioritization allows for a clearer contrast between regions with differing epidemiological trends. Data from other global regions have been retained selectively, serving to contextualize overarching patterns or to illustrate notable exceptions. We believe this revised structure enhances the clarity and relevance of the geographic comparison, and aligns more closely with the review’s central objective.
6.2 Culture
Please add: “false negative results can arise if patients have taken PPIs within 2 weeks or antibiotics within 4 weeks prior testing” – as stated in line 61 of subsection 6.4 Urea breath test.
Response: Thank you for highlighting the importance of pre-test medication history. In response, we have added a statement to Section 6.2 noting that false negative results may occur if patients have recently taken proton pump inhibitors (PPIs) or antibiotics, as described in subsection 6.4. To support this addition, we have also included the corresponding reference ( ref. 208) to ensure consistency and traceability across sections.
Please verify the number of references, ex. 232 in the text corresponds to 234 from the reference list, 233 in the text corresponds to 235 from the reference list, 239 in the text corresponds to 241 from the reference list ….
Response: Thank you for pointing out the discrepancies in reference numbering. We have reviewed the in-text citations and cross-checked them against the reference list. Corrections have been made to ensure that all citation numbers correspond accurately to the intended references. We appreciate your attention to detail and have verified consistency throughout the manuscript.
7.1 Antibiotic resistance
When referring to antibiotic resistance, please specify in every instance whether primary or secondary resistance is meant.
Response: The point is well taken. However, this information was not widely available and we did not have reliable information on this. We acknowledge this limitation in the discussion on antibiotic resistance,
Table 1. There are significant differences between Eastern and Western European countries; perhaps it would be a good idea to refer only to Western European countries, or, even better, to separate the first line of the table into Western Europe and Eastern Europe (studies from Eastern Europe should also be identified and included, for a comprehensive review).
Response: This is an important issue. However, unfortunately we did not have enough data from Eastern Europe to put it in to the Table
Conclusions – please provide a distinct conclusions paragraph, summarizing the main findings of this review.
Response: The point is well taken and we have indeed added a distinct conclusion paragraph.
Reviewer 2 Report
Comments and Suggestions for Authors
The authors provide a thorough and comprehensive review of the epidemiology, genetics, disease progression, and treatment of Helicobacter pylori infections, with particular focus on how these issues vary with regard to geographic regions. As such, the review addresses a gap in how we consider H. pylori infections worldwide. For example, the authors discuss how incidences of diseases related to H. pylori infections, such as gastric cancer, are disproportionately higher in East Asia, and how genetic variations in H. pylori impact this disparity. While this topic has been reviewed previously (e.g., Yamaoka Y, Kato M, Asaka M. Geographic differences in gastric cancer incidence can be explained by differences between Helicobacter pylori strains. Intern Med. 2008;47(12):1077-83; Zaidi SF. Helicobacter pylori associated Asian enigma: Does diet deserve distinction? World J Gastrointest Oncol. 2016 Apr 15;8(4):341-50), this review provides some more recent research results on the topic. This is a review that I would find valuable as a reference for my own research and teaching. The papers cited in the review are appropriate as are the conclusions drawn by the authors from the papers. The manuscript is well written for the most part and easy to read. I just have the following minor comments on the manuscript.
- Remove the second “75%” on line 92.
- The authors sometimes uses “USA” and other places “United States”. Either term is okay (I would prefer United States though), but be consistent in whichever term is used throughout the paper.
- Italicize H. pylori on line 107, Helicobacter on line 508, and H on lines 874 and 876.
- Remove “J.” on line 123.
- Change 170.000 to 170,000 on line 127.
- No need to italicize iatrogenic on line 139.
- Include a period (.) after the references indicated in line 199.
- The sentence in lines 204 to 208 is a run-on and is difficult to follow. The authors should consider revising it.
- Change the colon on line 240 to a period.
- Change the font size for the references on line 336.
- Including a list of abbreviations at the end of the review would be helpful as there are several abbreviations in the manuscript. Some of the abbreviations are unnecessary as they are only used once or twice in the manuscript. For example, there is no need to abbreviate peptic ulcer disease on line 337 since the abbreviation is only used once thereafter (on line 725). Similarly, there is no need to use the abbreviation ME-NBI on line 431, MTB on line 524, of VPZ on line 835.
- Include a space before the reference on lines 392, 408, 433, 809, 871.
- Remove period after “samples” on line 548.
- It is not clear why several phrases in lines 556-573 are in boldface. The highlighting of the phrases is distracting and should be changed to regular font style.
- Change Helicobacter pylori to H. pylori on line 775.
- Change H. Pylori to H. pylori on lines 805, 807, and 810.
- "Mucosa" is misspelled on line 811.
- It is unclear what the term ”bid” indicates with regard to antibiotic doses – e.g., “vonoprazan 20mg bid” on line 913 is the first place it appears. The authors appear to explain the term on line 909, but it is unclear what the term means from that explanation.
Author Response
Remove the second “75%” on line 92.
Response: Thank you for pointing this out. This has been undertaken
The authors sometimes uses “USA” and other places “United States”. Either term is okay (I would prefer United States though), but be consistent in whichever term is used throughout the paper.
Response: We agree and use US throughout the review.
Italicize H. pylori on line 107, Helicobacter on line 508, and H on lines 874 and 876.
Response: Done
Remove “J.” on line 123.
Response: Done
Change 170.000 to 170,000 on line 127.
Response: Done
No need to italicize iatrogenic on line 139.
Response: Changed
Include a period (.) after the references indicated in line 1
Response: Unfortunately, we do not understand this comment.
The sentence in lines 204 to 208 is a run-on and is difficult to follow. The authors should consider revising it.
Response: We agree and have split this long sentence into two sentences, amking it easier to read.
Change the colon on line 240 to a period.
Response: Done
Change the font size for the references on line 336.
Response: Done
Including a list of abbreviations at the end of the review would be helpful as there are several abbreviations in the manuscript. Some of the abbreviations are unnecessary as they are only used once or twice in the manuscript. For example, there is no need to abbreviate peptic ulcer disease on line 337 since the abbreviation is only used once thereafter (on line 725). Similarly, there is no need to use the abbreviation ME-NBI on line 431, MTB on line 524, of VPZ on line 835.
Response: This has been done.
Include a space before the reference on lines 392, 408, 433, 809, 871.
Response: Done
Remove period after “samples” on line 548.
Response: Done
It is not clear why several phrases in lines 556-573 are in boldface. The highlighting of the phrases is distracting and should be changed to regular font style.
Response: This has been changed to regular fonts
Change Helicobacter pylori to H. pylori on line 775.
Response: Corrected
Change H. Pylori to H. pylori on lines 805, 807, and 810.
Response: Corrected
"Mucosa" is misspelled on line 811.
Response: Corrected
It is unclear what the term ”bid” indicates with regard to antibiotic doses – e.g., “vonoprazan 20mg bid” on line 913 is the first place it appears. The authors appear to explain the term on line 909, but it is unclear what the term means from that explanation.
Response: Has been corrected
Reviewer 3 Report
Comments and Suggestions for Authors
1. Introduction
1) It is better to mention the out-of-africa story at the beginning several sentences of the review paper: Falush D, et al. Traces of human migrations in Helicobacter pylori populations. Science. 2003;299:1582–1585. doi: 10.1126/science.1080857.
2) For H. ylori severity in east asia, also provide some possible explanantions with corresponding references, not just description of the severe situation. Except for the virulent strains, is it possible for any specific lifestyles or food that may exacerabate the situation in east asia?
2. Epidemiology and prevalence
1) Why the high prevalence of H. pylori in Africa but very low gastric cancer rate? worth discussing in the manuscript. Line 79
2) The second paragraph of section 2 should accompanied by a global map showing the latest epidemiology of H. pylori infection.
3) Line 100: globa prevalence is dropping. It is true and a recent national study showing that the infection rate of H. pylori in Chinese urban population is 27.08%. Please have a look and consider citing it in the third paragraph of section 2 (line 101-128). https://pubmed.ncbi.nlm.nih.gov/38437848/ Multicentre, cross-sectional surveillance of Helicobacter pylori prevalence and antibiotic resistance to clarithromycin and levofloxacin in urban China using the string test coupled with quantitative PCR
3)
3. Transmission and re-infection
1) Please clarify the identification of H. pylro in oral and water environment. Whether only nucleic acids are detected or live bacteria were isolated and cultured? Make sure these are mentioned in sections 3.3 and 3.4 by citing relevant references.
2) Food transmission. H. pylori is very fragile. How long does H. pylori can survive in the environment? Is there any study showing that H. pylori can be transmitted via the food supply?
3) Any study reporting H. pylori transmission between humans and animals? Please clarify.
4) Section 3.7. Make sure to mention the clinical meaning of the low recurrence rate after H. pylori eradication. Please refer to certain published papers for the potential clinical meaning of the low recurrence rate.
4. Genetic diversity and population structure
The overall disadvantage of this part is the lack of in-depth description, just stacking facts. For example, for the genomic plasticity of H. pylori, how plastic is that? How do researchers quantify it and evaluate it? For microevolution, what is the mutation rate of the genome? For virulence genes, except for vacA and Caga, any other virulence factors? These are also very important for an insightful reviewer paper. Please do not just list all the facts that H. pylori researchers know very well. Otherwise, it is more like a popular science chapter.
5. Disease burden and clinical patterns
1) 5.6 diagnostic approaches and treatment strategies. Since section 6 focuses on diagnostic approaches, I would not expect the appearance of any diagnostic approaches here.
2) mention Correa's cascade, not just gastritis and gastric cancer.
6. Diagnostic approaches
1) Please refer to the latest review paper about H. pylori diagnosis by Prof. Barry Marshall's group. https://pubmed.ncbi.nlm.nih.gov/38910506/ Rapid diagnosis and precision treatment of Helicobacter pylori infection in clinical settings
2) String test plus qPCR methods should be mentioned by referring to the latest studies. THe current reference is too odd.
7. Antibiotic resistance and treatment strategies
1) A figure or table showing antibiotic resistance profiles and patterns is recommended.
2) Treatment strategies should be compared not just geograhically, but also effectively. What strategies should be used for which situation?
8. Line 929: "In summary, the H. pylori, eradication..." should be "for H. pylori". Otherwise, the sentence looks odd. Line 934 "second line tailored", what does this mean?
Other comments
1) It is suggested that several tables and figures should be added to the review paper in order to improve the illustration of the descriptions, such as epidemiology, diagnostic methods, and antibiotic resistance, etc. Currently, no figure and only one table are included in the manuscript, which is not sufficient for a full-length review article.
2) References need to be updated, avoiding too old ones except for that they are classical.
3) language needs to be thoroughly checked for grammar mistakes and typos.
Author Response
- Introduction
1) It is better to mention the out-of-africa story at the beginning several sentences of the review paper: Falush D, et al. Traces of human migrations in Helicobacter pylori populations. Science. 2003;299:1582–1585. doi: 10.1126/science.1080857.
Response: This reference has been added
2) For H. pylori severity in east asia, also provide some possible explanantions with corresponding references, not just description of the severe situation. Except for the virulent strains, is it possible for any specific lifestyles or food that may exacerabate the situation in east asia?
Response: Unfortunately we are not aware of any explanation in terms of life style or diet that might explain the severity of H. pylori in the East.
- Epidemiology and prevalence
1) Why the high prevalence of H. pylori in Africa but very low gastric cancer rate? worth discussing in the manuscript. Line 79
Response: This is an excellent point and is discussed in paragraph 5. 2 on gastric cancer.
2) The second paragraph of section 2 should accompanied by a global map showing the latest epidemiology of H. pylori infection.
Response: We find this an extremely complicated to design a global map to show the latest epidemiology of H. pylori, but would like the Editor of the journal decide whether this is necessary.
3) Line 100: globa prevalence is dropping. It is true and a recent national study showing that the infection rate of H. pylori in Chinese urban population is 27.08%. Please have a look and consider citing it in the third paragraph of section 2 (line 101-128).
https://pubmed.ncbi.nlm.nih.gov/38437848/ Multicentre, cross-sectional surveillance of Helicobacter pylori prevalence and antibiotic resistance to clarithromycin and levofloxacin in urban China using the string test coupled with quantitative PCR
Response: This reference has been added.
- Transmission and re-infection
1) Please clarify the identification of H. pylro in oral and water environment. Whether only nucleic acids are detected or live bacteria were isolated and cultured? Make sure these are mentioned in sections 3.3 and 3.4 by citing relevant references.
Response: This has been mentioned as suggested.
2) Food transmission. H. pylori is very fragile. How long does H. pylori can survive in the environment? Is there any study showing that H. pylori can be transmitted via the food supply?
Response: This has been mentioned as suggested.
3) Any study reporting H. pylori transmission between humans and animals? Please clarify.
Response: This has been mentioned as suggested.
4) Section 3.7. Make sure to mention the clinical meaning of the low recurrence rate after H. pylori eradication. Please refer to certain published papers for the potential clinical meaning of the low recurrence rate.
Response: This has been mentioned as suggested.
Genetic diversity and population structure
The overall disadvantage of this part is the lack of in-depth description, just stacking facts. For example, for the genomic plasticity of H. pylori, how plastic is that? How do researchers quantify it and evaluate it? For microevolution, what is the mutation rate of the genome? For virulence genes, except for vacA and Caga, any other virulence factors? These are also very important for an insightful reviewer paper. Please do not just list all the facts that H. pylori researchers know very well. Otherwise, it is more like a popular science chapter.
Response:.
Thank you very much for your insightful comments. We fully agree with your assessment—the section originally lacked sufficient analytical depth, and your suggestions were instrumental in guiding a more rigorous and informative revision. In response, we’ve substantially revised the Microevolution and Genomic Plasticity subsection, incorporating new references and detailed data from intra-host comparisons, longitudinal sampling, and sequential isolate analyses. These additions serve to illustrate H. pylori’s exceptional genomic flexibility—quantified through mutation rates and recombination metrics—and to clarify how such plasticity underpins its persistence, host adaptation, and resistance development. Accordingly, additional revisions were made across Section 4 to improve the overall clarity and relevance of the review.
Disease burden and clinical patterns
1) 5.6 diagnostic approaches and treatment strategies. Since section 6 focuses on diagnostic approaches, I would not expect the appearance of any diagnostic approaches here.
Response: The point is well taken but we think the diagnostic approach needs to be mentioned also in this part of the review.
2) mention Correa's cascade, not just gastritis and gastric cancer.
Response: This is a very good point and this has been has been added in the beginning of the section on gastric cancer, 5.2.
- Diagnostic approaches
1) Please refer to the latest review paper about H. pylori diagnosis by Prof. Barry Marshall's group. https://pubmed.ncbi.nlm.nih.gov/38910506/ Rapid diagnosis and precision treatment of Helicobacter pylori infection in clinical settings
Response: This has been added.
2) String test plus qPCR methods should be mentioned by referring to the latest studies. THe current reference is too odd.
Response: This has been added.
- Antibiotic resistance and treatment strategies
1) A figure or table showing antibiotic resistance profiles and patterns is recommended.
Response: We already have a Table on antibiotic resistance and perhaps figure is therefore not needed in our opinion.
2) Treatment strategies should be compared not just geograhically, but also effectively. What strategies should be used for which situation?
Response: Although this is an important question, we are not aware of global studies showing a major differences in efficacy other than based on antibiotic resistance which is related to the geographical regions.
- Line 929: "In summary, the H. pylori, eradication..." should be "for H. pylori". Otherwise, the sentence looks odd. Line 934 "second line tailored", what does this mean?
Response: We agree and this has changed as suggested, the word tailored has been removed.
Other comments
1) It is suggested that several tables and figures should be added to the review paper in order to improve the illustration of the descriptions, such as epidemiology, diagnostic methods, and antibiotic resistance, etc. Currently, no figure and only one table are included in the manuscript, which is not sufficient for a full-length review article.
Response: We have added a figure on transmission and diagnostic methods
2) References need to be updated, avoiding too old ones except for that they are classical.
Response: Some old references have been replaced by
3) language needs to be thoroughly checked for grammar mistakes and typos.
Response: This has been undertaken.
Round 2
Reviewer 1 Report
Comments and Suggestions for Authors
The authors responded to my earlier remarks and performed some of the necessary changes.
However, a few changes are still needed.
Table 1
Please add to Culture, Disadvantages / Limitations: “false negatives with recent PPI/antibiotic use”
Regarding Table 2 (former Table 1 in the initial version), my remarks and the authors’ response were:
My initial remarks:
Table 1. There are significant differences between Eastern and Western European countries; perhaps it would be a good idea to refer only to Western European countries, or, even better, to separate the first line of the table into Western Europe and Eastern Europe (studies from Eastern Europe should also be identified and included, for a comprehensive review).
Authors’ response: This is an important issue. However, unfortunately we did not have enough data from Eastern Europe to put it in to the Table
Current suggestion for improvement:
Studies from Central and Eastern European countries have been performed and published. If omitting those countries was not by an assumed choice of the authors, articles from countries in Central and Eastern Europe should be identified and included in the review.
Otherwise, please assume this choice of omitting countries from Central and Eastern Europe and make this omission clear in Table 2, as well as in the Discussions of your manuscript.
In this case, the authors should explicitly mention both in the legend of Table 2 and in the manuscript that this review only addressed 3 Western European countries (Germany, France, UK) and excluded other European countries.
Similarly, North America should then be clearly defined as USA and Canada only (therefore, excluding Mexico by choice, once again, since studies have been performed and published from that North American country as well).
Please make sure that Helicobacter pylori or H pylori is written in italic font.
My initial remark:
7.1 Antibiotic resistance
When referring to antibiotic resistance, please specify in every instance whether primary or secondary resistance was meant.
Response: The point is well taken. However, this information was not widely available and we did not have reliable information on this. We acknowledge this limitation in the discussion on antibiotic resistance.
Current suggestion for improvement:
Many studies and reviews actually do offer a clear distinction between primary and secondary resistance. As this distinction has important clinical implications, this information should be identified and extracted from such studies and this distinction should be presented and discussed, possibly starting from 2 separate tables, each of them concerning one of these main types of resistance (primary vs. secondary).
Author Response
Comment 1, Add to Table 1. false negative with PPI/antibiotic use.
Response: The point is well taken and this has been added
Comment 2. On the lack of data from Eastern Europe
Response: The reviewer is right that it would be good to have as much data as possible from different countries. However, one of the main aim of the review is to compare West and East and there is unfortunately limited data from Eastern Europe.
We have modified the Table 2 and explained that this is Western Europe and the countries, as well as North America, being USA and Canada. We have also acknowledged this as a limitation of this part of the review not being able to include Eastern countries
Reviewer 3 Report
Comments and Suggestions for Authors
Ref. 175 Format error. Please correct it.
Author Response
No response was requested
Round 3
Reviewer 1 Report
Comments and Suggestions for Authors
The authors responded to my earlier remarks and performed the necessary changes.